# Cataract Surgery following Corneal Allogenic Intrastromal Ring Segments and Implantable Collamer Lens

**Arsalan Akbar Ali** [1], **Bobby Saenz** [2] and **Taj Nasser** [2,*]

1   Anne Burnett Marion School of Medicine, Fort Worth, TX 76109, USA; arsalan.ali@tcu.edu
2   Parkhurst NuVision, San Antonio, TX 78229, USA
*   Correspondence: tajnassermd@gmail.com

**Abstract:** The management of cataracts in keratoconus patients poses a challenge due to the irregular corneal shape and variability in corneal topography, which may lead to errors in determining corneal power. In this report, we present a case of a 48-year-old male with a history of keratoconus and prior Visian Implantable Collamer lenses and Corneal Allogenic Intrastromal Ring Segments procedures, who presented with a nuclear cataract in his right eye. To address this patient's complex case, he underwent ICL explantation, cataract extraction, and intraocular lens (IOL) implantation, utilizing the Johnson & Johnson Sensar AR40 monofocal 3-piece lens with a power of $-9.5$. The Barrett True K formula predicted a spherical equivalent of $-1.76$, and at the post-operative one-month follow-up, the uncorrected distance visual acuity (UDVA) was 20/60, with pinhole improvement to 20/50. The manifest refraction was $-2.50$–$3.25 \times 145$, and the best corrected visual acuity was 20/25. This case report highlights the unique challenges encountered in managing keratoconus patients with a history of prior ICL and CAIRS procedures, followed by cataract extraction. Our findings underscore the importance of a comprehensive approach in the management of progressive keratoconus and cataracts to ensure optimal outcomes.

**Keywords:** keratoconus; CAIRS; cataract; surgery

## 1. Introduction

Keratoconus is a degenerative disorder of the cornea that leads to its thinning and progressive steepening, resulting in irregular astigmatism and visual impairment [1]. Various treatment modalities have been developed to improve the quality of vision and potentially halt the progression of keratoconus in patients. Treatment options include using rigid contact lenses to reshape the cornea and correct astigmatism, corneal collagen cross-linking and intrastromal corneal ring segment implantation to stabilize the cornea, implantable collamer lenses to decrease refractive error, and in advanced cases, surgery such as anterior lamellar or penetrating keratoplasty [2]. CAIRS (Corneal Allogenic Intrastromal Ring Segments) is a novel surgical technique for keratoconus patients that was recently developed and was found to be effective in a pilot study [3]. Implantable collamer lenses (ICL) present as a feasible refractive option in keratoconus patients given corneal laser resurfacing is obviated.

A significant proportion of patients with keratoconus may develop cataracts, leading to further deterioration of their vision. Managing patients with progressive keratoconus and cataracts requires a multifaceted approach as the corneal irregularity can confound and make it difficult to accurately calculate the intraocular lens calculations [4]. Corneal asymmetry and variations in corneal topography can lead to inaccuracies in determining corneal power, which has a direct impact on the final lens power parameters. For the best visual quality, it is crucial to address corneal irregularity and flatten keratometric values before attempting cataract surgery and intraocular lens implantation in order to ensure the most accurate outcome [5].

To the best of our knowledge, there have been no reports of a patient with history of ICL and CAIRS to subsequently undergo cataract extraction, highlighting the unique nature of this case.

## 2. Case Report

A 48-year-old male presented to our clinic with a nuclear cataract in his right eye. The patient has a history of keratoconus and had previously undergone Visian ICL surgery in order to have a pair of glasses that would allow him to not have to wear scleral lenses all the time. After the ICL surgery, the scleral lenses were still causing irritation and the patient wanted to try another surgical option without pursuing a penetrating keratoplasty. The patient subsequently underwent CAIRS in the same eye to allow for improved fitting of scleral lenses and visual outcome. Approximately three weeks after the CAIRS procedure, the patient complained of progressively worsening blurry vision and desired an evaluation to maximize quality of vision.

The initial physical examination of the right eye revealed a visual acuity of 20/400 with pinhole improvement to 20/70, intraocular pressure of 10 mmHg, and round and reactive pupils without relative afferent pupillary defect. External examination was unremarkable. On slit-lamp examination (SLE), the anterior segment of the eye revealed mild anterior stromal thinning, allogenic intrastromal ring segments well placed in the mid-peripheral cornea, and a deep and quiet anterior chamber (Figure S1). The lens revealed 3+ nuclear sclerosis and the Visian ICL was well positioned with a vault of approximately 100% central corneal thickness. The posterior segment was unremarkable. The left eye was unremarkable with a best corrected visual acuity of 20/20.

Diagnostic modalities included an Oculus Pentacam which reported the keratometry (K) indices for the flat and steep axis OD were 54.11 diopters (D) at 105° and 54.65 D at 15° respectively (Figure 1). Anterior Segment Optical Coherence Tomography highlighted an ICL vault of 518 μm (Figure 2). Biometry was performed using the Pentacam AXL, and the aim was to achieve a spherical equivalent of approximately −1.75 to −2.00 to avoid hyperopic results due to the patient's severe keratoconus. A Johnson & Johnson Sensar AR40 monofocal 3-piece lens with a power of −9.5 with a predicted spherical equivalent of −1.76 using Barrett True K formula. Femtosecond laser use was attempted to perform parts of the surgery including capsulorhexis formation however this was unable to be completed due to significant scarring of the cornea, as well as CAI impeding adequate view. A 2.75 mm incision was made temporally and the ICL was extracted taking care to avoid damage to the cornea. Generous use of dispersive viscoelastic was helpful in ensuring optimal view throughout the procedure. Of note, the view was difficult thus proper magnification and lighting was ensured to aid with capsulorhexis formation. Nucleus disassembly was performed using a modified stop-and-chop technique and following cortex removal, the three-piece lens was injected using standard technique. Intracameral antibiotic and steroid were injected, and incisions were ensured to be watertight at the conclusion of the case.

Outcome at postoperative day 1 (POD1) revealed an uncorrected distance visual acuity (UDVA) of 20/70 + 1 with pinhole improvement to 20/60, intraocular pressure of 8 mmHg, mild corneal edema temporally at the incision site, and incisions were Seidel negative. The anterior chamber had 1+ cell and trace flare and the monofocal IOL was well positioned in the bag. Postoperative week 1 (POW1) follow-up demonstrated an improved UDVA of 20/50 with pinhole improvement to 20/30. The intraocular pressure remained at 8 mmHg. Postoperative month 1 (POM1) follow-up, patient was found to have UDVA of 20/60 with pinhole improvement to 20/50. The intraocular pressure was found to be 8 mmHg. Manifest refraction at this visit was −2.50–3.25 × 145 with the best corrected visual acuity of 20/25 with glasses. The patient was satisfied with his results as the goal was to improve functional vision and quality of life.

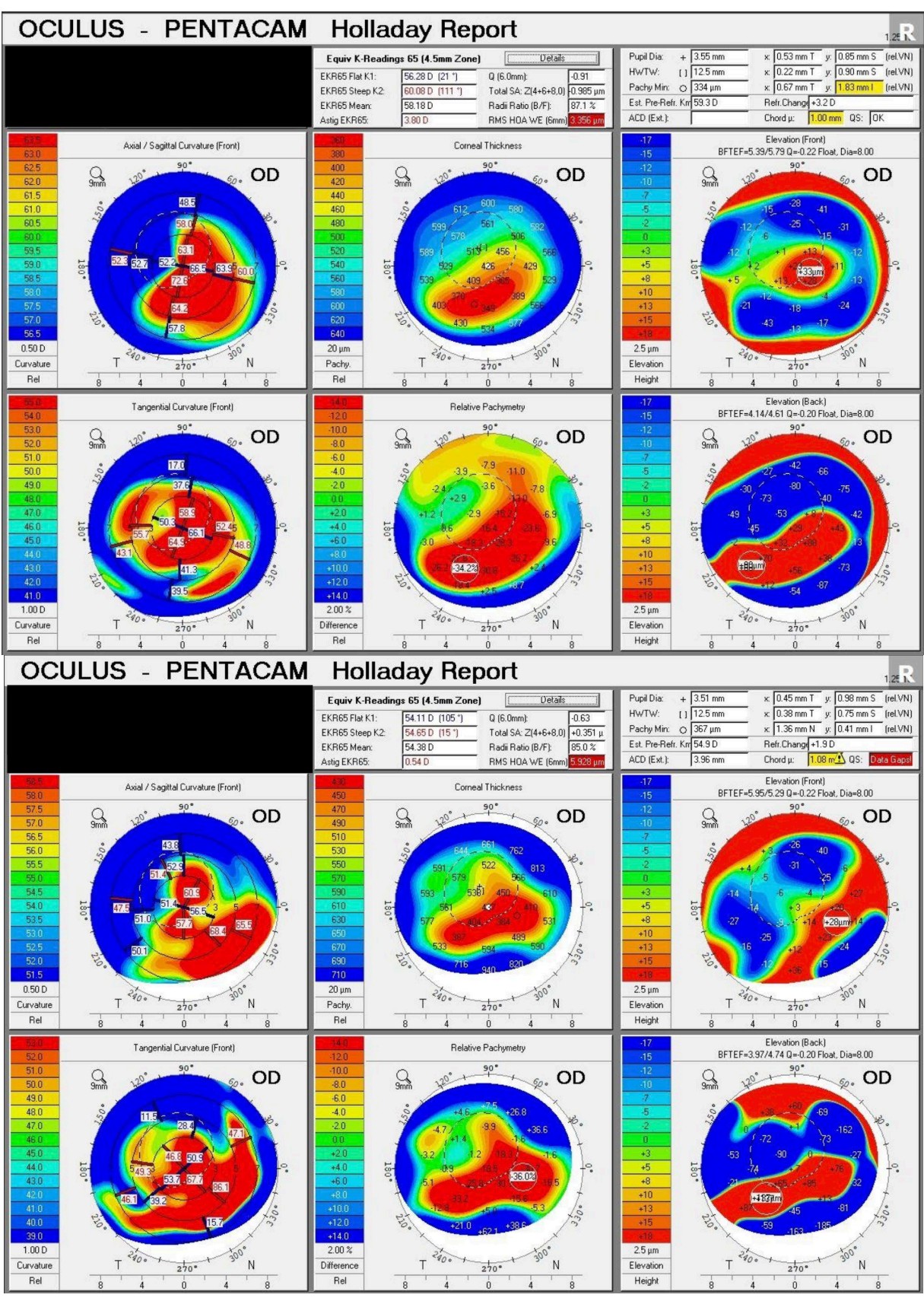

**Figure 1.** Sheimpflug tomography demonstrates severe keratoconus as the image displays significant steepening of the cornea before and after the CAIRS procedure. We can appreciate the 10 D flattening of the central cornea that occurred after the surgery.

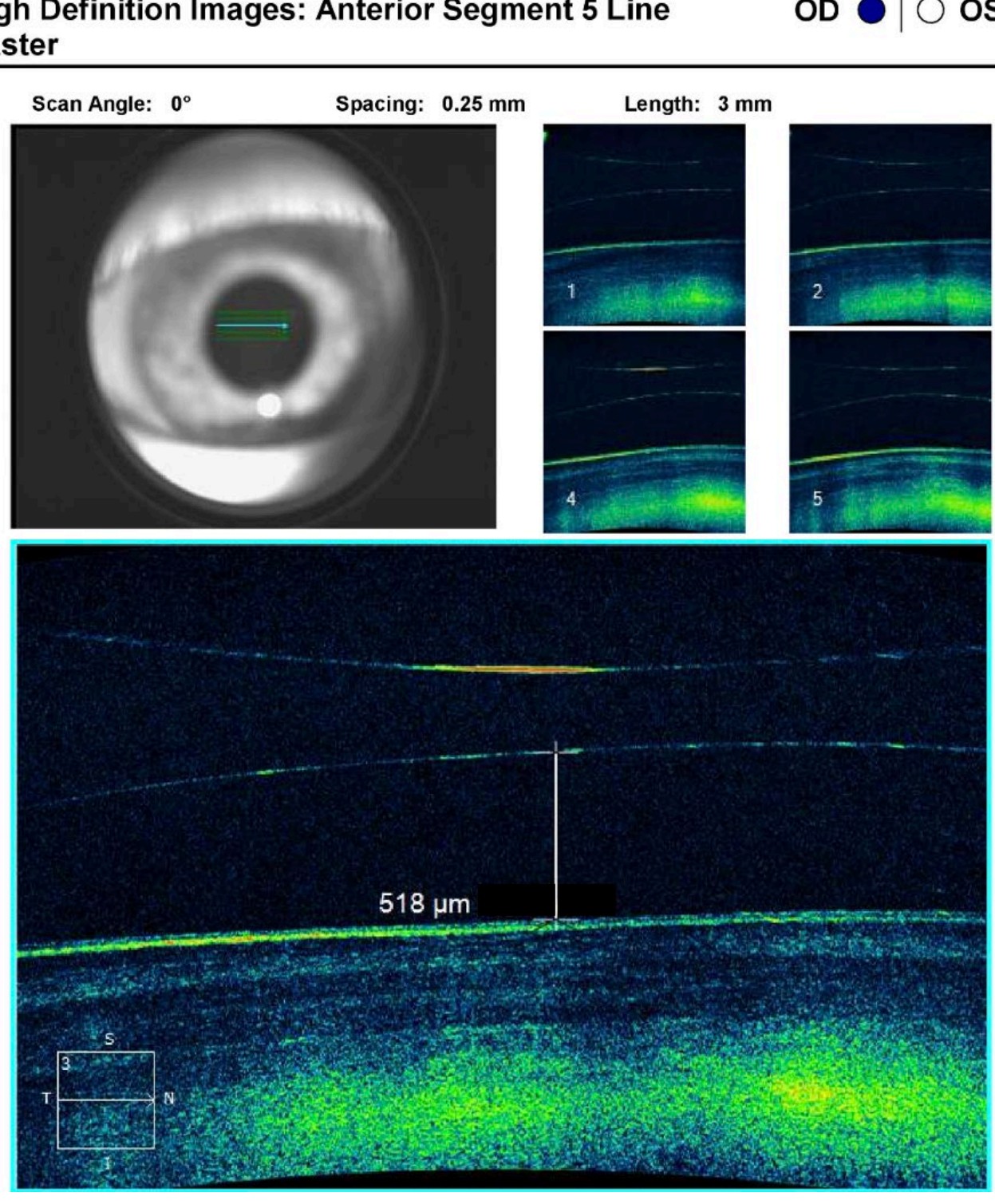

**Figure 2.** Anterior segment optical coherence tomography measuring a normal vault, which is the distance between posterior surface of the ICL and anterior lens capsule.

### 3. Discussion

Managing cataracts in patients with keratoconus can present a range of challenges, making it essential to have thorough preoperative planning for successful surgical outcomes. The irregular shape of the cornea in keratoconus can considerably affect the accuracy of standard measurements, making it difficult to calculate the correct intraocular lens (IOL)

power for cataract surgery [6,7]. The parameters of IOL calculation that must be considered to provide the best possible outcomes include keratometry, axial length, effective lens position, degree of astigmatism, and ocular surface [8]. Currently, all biometry formulas available require axial length and keratometry value (K)_as minimal parameters to evaluate refractive power [9]. Accurate measurements of the axial length is important as keratoconus patients often have deeper anterior chambers and have been observed to have longer axial lengths when compared to individuals without the condition [10]. Additionally, the apex of the cornea is not centrally located, which can lead to unreliable measurement of the visual axis [10]. Several studies have depicted that the more advanced the keratoconus, the less repeatable the keratometry values [9,11]. The unstable tear film over the cones adds to the complexity as often there are variations with sets of calculations within hours of each other [12]. Our team countered some of the challenges by ensuring devices were calibrated, optimizing the surface, and repeating corneal topographic scans and biometry to confirm repeatability. Moreover, due to risk the of hyperopic error post-operatively, we chose to aim for a myopic spherical equivalent. The work of Watson et al. supported this as the author suggested that when selecting an intraocular lens (IOL) for patients with mild and moderate keratoconus (mean K $\leq$ 55 D), utilizing the individual's actual keratometry values and aiming for a low level of myopia is an appropriate approach [13]. On the other hand, in cases of severe keratoconus (mean K > 55 D), using the patient's actual K values may lead to a significant hyperopic error. Therefore, an alternative option to consider in these cases would be using a standard K value for IOL selection [13].

Although the fundamental steps of cataract removal and IOL implantation remain consistent with the standard procedure, certain technical modifications intraoperatively can improve patient outcomes. The case presented was unique as the patient had an intraocular collamer lens (ICL) implant to debulk his myopia. ICL surgery may not be a common treatment option for keratoconus, but it can be considered for patients with high levels of myopia that cannot be corrected with glasses or contact lenses [14,15]. This surgery can help to improve visual acuity by reducing refractive error. However, it is important to note that ICL surgery does not cure keratoconus or address its underlying corneal thinning and bulging. Patients who undergo ICL surgery for keratoconus should continue to manage their condition with specialized contact lenses or other treatments. In this case, the ICL implant was explanted prior to cataract extraction. The procedure was performed easily due to the flexible collamer material of the ICL. Care was taken to grasp the central portion of the optic during removal in order to avoid damage to the fragile footplates and facilitate removal in one piece. As in modern cataract surgery, the main corneal incision is a critical step. However, in keratoconus eyes caution must be taken to use the standard approach as it may lead to post-operative K values and corneal shape change in an unpredictable manner [9]. The amount of corneal scarring present should also be considered when planning the location of the incision [9]. Capsulorrhexis can be challenging when the cornea is scarred and irregular, as it results in light scattering which can cause poor intraocular visualization for the surgeon [16]. The scattering of light also causes a loss of contrast between the tissue, which can further decrease the surgeon's depth perception. Additionally, the irregular shape of the cornea can also make it difficult to accurately position the intraocular lens. Three options can aid the surgeon in performing capsulorrhexis in keratoconus patients: using trypan blue to stain the anterior capsule, injecting a dispersive ophthalmic viscoelastic device (OVD) such as hydroxypropyl methylcellulose 2% gel onto the corneal surface for greater hydration and magnification, or a newer experimental option of injecting a cohesive OVD onto the corneal surface and placing an RGP contact lens on top for greater visibility and less image distortion with no complications [17,18].

Postoperatively, the management of patients undergoing cataract extraction in keratoconus is no different from the norm as it involves monitoring for complications [19]. However, a limitation many studies have shown is that residual refractive error may be common in keratoconus patients especially those with advanced disease [16,20,21].

Refraction, which measures the eye's ability to focus light on the retina, is crucial in determining a patient's eyeglass prescription. However, the irregular shape of the cornea in keratoconus patients can cause inaccurate refraction measurements. Determining the appropriate intraocular lens poses difficulties, primarily due to the irregular corneal shape, which scatters light rays in various directions, leading to blurred vision. Furthermore, the progressive nature of the disease complicates refraction measurements even further.

## 4. Conclusions

In conclusion, this case report describes a unique situation of a patient with a history of keratoconus who underwent corneal collagen crosslinking, Visian ICL surgery, and CAIRS, subsequently requiring cataract extraction and IOL implantation. The presence of progressive keratoconus and cataracts requires a multidimensional approach as corneal irregularity can complicate and confound intraocular lens calculations. Careful preoperative measurements, including topography and tomography, are necessary to map out the corneal surface and identify any areas of steepening or irregularity. Based on these measurements, the refractive target is calculated and adjusted to ensure the best possible visual outcome for the patient. Moreover, repeated scans are also crucial in monitoring any changes in corneal shape and adjusting the IOL power accordingly. Through the utilization of a comprehensive and multifaceted approach, we successfully achieved improved visual acuity and overall satisfaction for the patient, resulting in a substantial improvement in their quality of life.

**Supplementary Materials:** The following supporting information can be downloaded at: https://www.mdpi.com/article/10.3390/jcto1030009/s1, Figure S1: Slit lamp findings.

**Author Contributions:** Conceptualization, T.N. and B.S.; methodology, T.N.; software, T.N.; validation, T.N. and B.S.; formal analysis, T.N.; investigation, T.N.; resources, T.N. and B.S.; data curation, A.A.A.; writing—original draft preparation, A.A.A.; writing—review and editing, A.A.A., T.N. and B.S.; visualization, T.N.; supervision, T.N.; project administration, T.N.; funding acquisition, T.N. All authors have read and agreed to the published version of the manuscript.

**Funding:** This research received no external funding.

**Institutional Review Board Statement:** Ethical review and approval were waived for this study due by by the TCU [Office of Research Compliance (ORC) on 3 March 2023/Institutional Review Board (IRB) Chair and/or IRB Chair's designee(s) on 3 March 2023/Institutional Review Board (IRB) at the Committee meeting] and it was determined that the activities described do not meet the regulatory definition of human subjects research.

**Informed Consent Statement:** Patient consent was waived due to the retrospective nature of the project.

**Data Availability Statement:** Not applicable.

**Conflicts of Interest:** The authors declare no conflict of interest.

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
