# Peer review of "Cataract Surgery following Corneal Allogenic Intrastromal Ring Segments and Implantable Collamer Lens"

_2813-1053, doi:10.3390/jcto1030009_

Round 1

Reviewer 1 Report

My comments

1.       ICL was extracted, and MF IOl was inserted- in patients with KC- this topic is still controversial. https://pubmed.ncbi.nlm.nih.gov/24702581/ any new data in this field would be beneficial.

2.        How will this patient enjoy being glasses-free with MF with an RX of -2.50 -3.25 x 145?

3.       How could it be that he got ICL for keratoconus? This does not make sense to me.    

4.       What was his close-up vision?

5.       20/60 with Rx of 2.50 -3.25 x 145? It seems something is wrong here.

6.       Figure 1 can you describe the image in more detail?

7.       Why do you present figure 2?

8.       Can you provide a slit lamp photo of his cornea?

9.       What is WTW-this determines the ICL size. https://www.ncbi.nlm.nih.gov/pmc/articles/PMC8589920

10.   Please explain how reading this care would improve patient care.

11.   Is the bcva with glass or contact lenses?

12.   Can you provide the video of the ICL removal?

13.   Not sure the discussion about the rhexis is relevant- please consider

14.   When did he have CXL?

15.   What about his other eye? What is the vision there?

16.   How could you select MF for this case, as you were unsure about the accuracy of the measurement? MF usually works if the patient has Plano vision.

17.   Is the patient happy?  How is his daily life activity- you could report on his NEI VFQ-25 results

18; please review the literature on the use of MF in the KC case. You could also run a survey on this topic (the survey is the only option for future study)

Reviewer 2 Report

Interesting cases.  Need to descried the surgical procedures detail, intraoperative  dificulties, clar results and conclusions

Reviewer 3 Report

Interesting case report.

Please give your opinion about cataract development 3 weeks after CAIRS, it is rather unusual as it is basically surface procedure and ICL was well positioned. How long ICL was in the eye?

Round 2

Reviewer 1 Report

1.       Yes, sorry. Why did you use a 3-piece lens?

2.       OK- please add this to the paper. About more functional living

3.       OK, please add this to the discussion. About debulking of myopia

4.       Please report on the close-up reading – if needed, re-examine the patient

5.       This is because, in advanced KC, the refraction is not accurate. Please mention this in the limitation. Also, add some limitations from here https://www.ncbi.nlm.nih.gov/pmc/articles/PMC5748308/

6.       Also, add 10 D of flatting

7.       OK. What was the patient vault in comparison?

8.       Please add this maybe as a supplement.

9.       Yes, but how did you choose the ICL? The wrong ICL size would lead to a cataract; thus, the reader must know how to choose the right ICL.

10.   OK. please also add the risk of performing cataract surgery on  a young myopic patient with no PVD (increased risk for RD  ~10%)

https://pubmed.ncbi.nlm.nih.gov/10599657/

11. OK

12. OK. This is a case report

13. OK. Unfortunately, you do not provide the video to assist the reader

14. OK

15. so he has 20\20 on this one eye! He probably can function well with one eye, I suppose. Please explain why do so much intervention in the other eye. It will never be as good as the other eye

16. sorry

17 . what were the reasons for surgery if the patient has 20\20 in one eye? Please perform the questionnaire. Usually, we need to see the questionnaire before and after the intervention. With one eye 20\20, he probably had no visual function problems. Was the patient not pleased before the surgery? What were his complaints?

Reviewer 2 Report

interesting cases! for discussions need more study refferences!

Conclussion can be improve
